# FERONIA Receptor Kinase Integrates with Hormone Signaling to Regulate Plant Growth, Development, and Responses to Environmental Stimuli

**DOI:** 10.3390/ijms23073730

**Published:** 2022-03-29

**Authors:** Yinhuan Xie, Ping Sun, Zhaoyang Li, Fujun Zhang, Chunxiang You, Zhenlu Zhang

**Affiliations:** 1State Key Laboratory of Crop Biology, College of Horticulture Science and Engineering, Shandong Agricultural University, Tai’an 271000, China; 15505483595@163.com (Y.X.); sunping_89@126.com (P.S.); zyli_1106@163.com (Z.L.); 17863805279@163.com (F.Z.); 2Department of Horticulture, College of Agriculture, Shihezi University, Shihezi 832003, China

**Keywords:** receptor-like kinase, FERONIA, phytohormone, growth and development, plant immunity

## Abstract

Plant hormones are critical chemicals that participate in almost all aspects of plant life by triggering cellular response cascades. FERONIA is one of the most well studied members in the subfamily of *Catharanthus roseus* receptor-like kinase1-like (*Cr*RLK1Ls) hormones. It has been proved to be involved in many different processes with the discovery of its ligands, interacting partners, and downstream signaling components. A growing body of evidence shows that FERONIA serves as a hub to integrate inter- and intracellular signals in response to internal and external cues. Here, we summarize the recent advances of FERONIA in regulating plant growth, development, and immunity through interactions with multiple plant hormone signaling pathways.

## 1. Introduction

As sessile species, plants constantly encounter complex and unfavorable environments that threaten their survival. Therefore, plants use receptors to sense extracellular environmental signals and induce intracellular physiological and biochemical responses via signal transduction cascades. Plasma membrane-localized receptor-like kinases (RLKs) function as a ‘bridge’ during signal transduction. They use an extracellular domain featuring ligand-binding sites to perceive different signal molecules. Upon binding ligands, the conformation of RLKs is changed, which subsequently leads the intracellular kinase domain to phosphorylate cytoplasmic factors and activate downstream signaling pathways [1,2]. The first plant RLK was identified in maize about 30 years ago [3]. To date, RLKs have become one of the largest plant receptor families, with more than 600 members in the model plant *Arabidopsis thaliana* [4]. RLKs are functional in almost every aspect of the plant life cycle, including growth (cell elongation), reproduction (fertilization), hormone signaling, abiotic stress tolerance, and immunity [1,2,5].

RLKs are classified into different types based on differences in the extracellular domain, such as proline-rich extensin-like RLKs, leucine-rich repeat RLKs, lectin RLKs, and malectin-like RLKs [1,5]. The malectin-like RLKs are also known as *Catharanthus roseus* receptor-like kinase 1-like (*Cr*RLK1Ls), named after the first *CrRLK1* isolated from the Madagascar periwinkle [6]. All members of the *Cr*RLK1L subfamily share conserved protein structures, including an N-terminal extracellular malectin-like domain, a middle transmembrane domain, and a C-terminal intracellular serine (Ser) and threonine (Thr) kinase domain [1,5]. Among the 17 members of the *Cr*RLK1L subfamily in *Arabidopsis*, FERONIA (FER), named after the Etruscan goddess of fertility, was initially identified to function in mediating male–female interactions during pollen tube reception [7,8,9,10,11]. In normal flowering plants, the penetrating pollen tube with sperms is induced to rupture by the female gametophyte; the sperm is then released into the female gametophyte for fertilization. However, in the *Arabidopsis feronia* (*fer*) mutant, the pollen tube fails to arrest and continues to grow inside the female gametophyte, resulting in impaired fertilization [7,8,9]. Further investigations show that the accumulation of reactive oxygen species (ROS, mainly composed of hydroxyl free radicals), regulated by the FER-RAC/ROP-NADPH oxidase-based signaling pathway, is responsible for inducing pollen tube rupture and sperm release [10,12]. The FER-regulated accumulation of ROS is also involved in mediating self-incompatibility in the Chinese cabbage (*Brassica rapa* L. ssp. *pekinensis*) [13]. More importantly, the latest findings demonstrate that FER functions by preventing the approach and penetration of late-arriving pollen tubes by triggering the accumulation of nitric oxide at the filiform apparatus upon the arrival of the first female gametophyte at the ovule [11]. After nearly two decades of research effort, FER has been found to play critical roles in cell growth, reproduction, plant morphogenesis, immunity, hormone signaling, and stress responses [1,5,14,15,16], suggesting its key roles in plant growth and development.

Phytohormones are critical plant chemicals that participate in diverse cellular and developmental processes at low concentrations. Phytohormones are involved in nearly all aspects of the plant life cycle, ranging from seed germination, growth and development, flowering, and fruit ripening, to immunity by integrating endogenous signals and extra-environmental cues. As a critical receptor kinase, FER is involved in many hormonal signaling pathways that regulate plant growth, development, and immunity [2,14,15]. Here, we focus on the recent findings related to the functions of FER in hormone signaling pathways and summarize them by different types of phytohormones.

## 2. Auxin

Auxin functions in nearly all aspects of plant growth and development [17,18], including root hair development. Root hair development also requires FER, as the *Arabidopsis fer* mutant shows severe root hair defects, such as collapsed, burst, and short root hairs [12]. Moreover, in a temperature-sensitive *feronia* mutant, which was derived from a G41S substitution in the extracellular domain, the root hair formation was compromised at elevated temperature (30 °C) [19]. The fact that *fer*-induced root hair defects cannot be rescued by exogenous auxin suggests that FER plays an indispensable role in auxin-mediated root hair growth [12,19].

Several pathways have been illustrated to explain the function of FER in auxin-regulated root hair growth. The first is that FER regulates RHO GTPase (RAC/ROP)-mediated NADPH oxidase-dependent ROS accumulation by interacting with the RHO GTPase guanine nucleotide exchange factor (ROPGEF) [12] (Figure 1, right panel). Specifically, RAC/ROP signaling is critical for polarized cell growth (e.g., root hair elongation) [20,21,22], and ROPGEF proteins preferentially bind to GDP-bound inactive RAC/ROP. Once activated, the GTP-bound activated RAC/ROP recruits NADPH oxidase to mediate downstream ROS production and ROS-dependent processes, such as the development of root hairs [12,23,24]. FER interacts with GTPase ROP2 and acts upstream of RAC/ROP signaling to regulate ROS accumulation and auxin-regulated root hair growth. However, the precise role of FER in this pathway has not been fully elucidated, including whether FER phosphorylates GTPase ROP2, and how FER affects the protein stability or function of GTPase ROP2. Loss of functional FER might disturb the balance between the inactive and activated forms of RAC/ROP, resulting in a significant decrease in ROS production mediated by the ROP GTPase pathway in root hairs [25,26]. Additionally, some other FER-related factors may be involved in this process. Li et al. reported that glycosylphosphatidylinositol-anchored protein (GPI-AP) LRE-like GPI-AP1 (LLG1) is a key component of the FER–RHO GTPase signaling complex [24]. Specifically, LLG1 interacts with and serves as a FER ‘chaperone’ that delivers FER from the endoplasmic reticulum (ER), where the interaction occurs, to the cytoplasmic membrane, where FER is functional [24]. Loss of LLG1 results in the retention of FER at the ER and a similar growth phenotype to that of the *fer* mutant, suggesting the crucial role of the LLG1–FER–ROPGEF–RAC/ROP signaling complex in ROS production and root hair growth [12,24].

However, FER-regulated root hair growth is not always RAC/ROP-dependent. Huang et al. reported that ROPGEF10 mainly contributes to initiating root hair development, while ROPGEF4 is only responsible for elongating root hairs [28]. Functional loss of ROPGEF4 or ROPGEF10 abolishes most FER-induced ROS production, suggesting that these two factors are important downstream components of FER-RAC/ROP signaling pathway [28]. Nevertheless, the initiation and elongation of root hair in single and double mutants of ROPGEF4 and ROPGEF10 were much easier than in the wild type (WT) when treated with 1-naphthaleneacetic acid [28]. These findings suggest that auxin-regulated root hair growth is independent of ROPGEF4 and ROPGEF10. A possible scenario is that FER interacts with and activates other cellular pathways in response to environmental cues. Thus, FER regulates root hair growth through ROPGEFs or other yet unknown factors.

The second possible pathway is that FER participates in auxin-regulated apoplastic pH to stimulate the expansion of root cells (Figure 1, left panel). Multiple reports have shown that apoplastic pH plays a critical role in regulating local cell expansion [29,30]. Barbez et al. further reported that reduced auxin levels, perception, or signaling severely affects the acidification of apoplasts and cellular expansion [31]. However, increasing exogenous and endogenous cellular auxin levels result in transient alkalization of the apoplast, leading to inhibited root cell elongation in a FER-dependent manner [31]. Interactions between FER and its ligand rapid alkalization factor 1 (RALF1) trigger phosphorylation of the plasma membrane-localized proton pump (H^+^-ATPase) [27]. Upon phosphorylation, the proton transport activity of H^+^-ATPase is inhibited, leading to subsequent alkalization of the apoplast [27]. It was further demonstrated that functional loss of FER abolishes auxin-induced alkalization of the apoplast, resulting in inhibited expansion of root cells [31]. Furthermore, ERULUS (ERU), a receptor kinase in the *Cr*RLK1L subfamily, is also involved in FER-regulated growth of root cells [32]. Specifically, auxin activates the expression of *ERU* through transcription factors auxin response factor 7 (ARF7) and ARF19, and also promotes phosphorylation of the ERU protein to regulate cell wall composition and cell elongation [32]. Loss of functional ERU results in decreased abundance of phosphorylated FER and an increased amount of phosphorylated H^+^-ATPase 1/2, which affects the apoplastic pH and results in growth defects in root hairs [27,32]. These findings confirm that FER is closely involved in auxin-mediated cell elongation by regulating apoplastic pH.

Additionally, FER is involved in polar auxin transport to regulate asymmetric root growth, and two independent research groups revealed that PIN-FORMED2 (PIN2) is a key factor in this process [33,34]. Specifically, in the *Arabidopsis fer* mutant, reduced cytoskeleton F-actin induces aberrant localization of PIN2, resulting in a delayed gravitropic response compared to that of the WT [33]. The other group found that suppressing polar auxin transport pharmacologically or by mutating *PIN2* or *AUX1* (*AUXIN RESISTANT1*) represses asymmetric root growth in the *fer* mutant [34]. These findings indicate a key role of FER in regulating asymmetric root growth via PIN2-mediated auxin transport.

## 3. Abscisic Acid

Originally identified as functional during plant abscission and senescence, abscisic acid (ABA) is commonly considered a cell growth restriction factor [35]. ABA is required for fine-tuned growth and development and plant responses to biotic and abiotic stressors, such as drought and salinity [36]. It is widely accepted that PYRABACTIN RESISTANCE (PYR), PYR1-LIKE (PYL), and REGULATORY COMPONENTS OF ABA RECEPTORS (RCAR) serve as ABA receptors [37,38,39,40]. ABA-bound PYR/PYL/RCAR proteins interact with A-type protein phosphatase 2C (PP2C) and inhibit its activity, which subsequently releases SNF1-related protein kinase 2 (SnRK2) in its active form [36,41,42]. SnRK2 is a critical positive regulator in the ABA signaling pathway and its activated form triggers ABA-responsive events, such as the activation of ABA response genes, stomatal closure, and the inhibition of cell growth [36,41,42].

Recent studies have revealed that FER is also involved in regulating ABA signaling by interacting with ABA INSENSITIVE 2 (ABI2), a PP2C member acting upstream in the ABA response [43,44]. The activated GTPase ROP11 preferentially binds FER-ROPGEFs, which then interact with and activate the phosphatase activity of ABI2, leading to the deactivation of SnRK2 and insensitivity to ABA [43] (Figure 2). ABA represses ABI2 activity via a receptor pathway to release SnRK2, which inhibits H^+^-ATPase2 (AHA2)-mediated acidification and root growth [42,45,46]. However, FER activates ABI2 activity through the ROPGEF–GTPase module to block the release of SnRK2, resulting in inhibited ABA responses, including cell growth. Therefore, any mutants in the FER–ROPGEFs–ROP11 signaling pathway are hypersensitive to ABA treatment [43]. However, ABI2 interacts with and promotes the dephosphorylation and deactivation of FER [44] (Figure 2). Therefore, FER and ABI2 form a loop to regulate each other’s activity and participate in ABA-mediated responses (Figure 2). Additionally, FaMRLK47, a FER paralog in strawberries (*Fragaria* × *ananassa*), interacts with FaABI1, a negative regulator of the ABA signaling, to modulate fruit ripening by regulating the ABA signaling pathways [47]. Overexpression of *FaMRLK47* causes a decrease in the expression of multiple ripening-related genes that are ABA-inducible [47]. Collectively, these findings indicate a pivotal role for FER in many ABA-mediated cellular biological processes.

The leucine-rich repeat (LRR)-extensins (LRXs) are a class of cell wall-localized chimeric extensin proteins [48]. Recent findings reveal that the N-terminal LRR domain of LRXs interacts with both FER and RALFs [49,50,51]. FER is a well-known receptor of RALF peptides [27,52,53,54], and the LRXs–RALFs–FER module has been demonstrated to play pivotal roles in regulating plant growth and immunity [49,50,51,55,56]. In the context of ABA responses, the ABA accumulation in *Arabidopsis* mutants *lrx345* and *fer-4* were dramatically increased, which contributed to the salt-hypersensitivity of these mutants [56]. Further investigation showed that the expression of *BG1* (β-glucosidase1), which is involved in converting the glucose-conjugated inactive form of ABA to the active form of ABA [57], was extensively increased in the *lrx345* mutant [56]. This indicates that the LRXs–RALFs–FER module regulates salt sensitivity in an ABA-dependent manner.

ROS is an important second messenger in the ABA signaling pathway [58]. The FER–ROPGEF–ROP–NADPH oxidase pathway mediated ROS production, which is involved in auxin-mediated growth of root hairs [12], also participates in the ABA signaling pathway [43]. ROS play a critical role in ABA-mediated stomatal closure [58]. Disrupting FER results in a smaller stomatal aperture than in the WT under ABA treatment (Figure 2), which may be caused by high ROS content in the *fer* mutant [43]. Thus, FER regulates ROS production through the GEF–ROP pathway to participate in both ABA- and auxin-mediated plant cell elongation. Additionally, the overaccumulated ROS in the mutants *fer-4* and aforementioned *lrx345* enhanced salt-induced cell death [56]. Blocking ABA biosynthesis by mutating *aba2* rescued the cell death phenotype in the *fer-4* and *lrx345* mutants under high salinity [56]. Further investigation showed that mutation of *aba2* in the two mutants did not affect the ROS accumulation but significantly downregulated expression of *RobhF* (*RESPIRATORY BURST OXIDASE HOMOLOG F*) [56], which encodes NADPH oxidase and is closely involved in ABA-promoted ROS production [59]. Moreover, ROS-responsive genes, including *ZAT10* (*SALT TOLERANCE ZINC FINGER 10*), *WRKY8*, and *SAG1* (*SENESCENCE ASSOCIATED GENE 1*)*,* were also attenuated in the *lrx345 aba2-1* mutants, especially under high salinity [56]. These results suggest that the *aba2* mutation rescued salt-induced cell death in the *fer-4* and *lrx345* mutants, and that this is at least partially dependent on the attenuated ROS responses under salt stress.

## 4. Brassinosteroids

Brassinosteroids (BRs) are a group of polyhydroxylated steroid phytohormones that were originally identified for their function in cell elongation [60,61,62]. BRs are crucial for diverse processes in plant growth and development, including cell division, senescence, vascular development, reproduction, and the response to biotic and abiotic stressors [62,63]. Tremendous progress has been made to decipher the signaling transduction pathways from cell surface receptors to the nucleus, where gene expression is modulated in response to BRs [63]. In brief, BRs bind to the receptor BRASSINOSTEROID INSENSITIVE1 (BRI1) on the cell surface and recruit the coreceptor BRI1-ASSOCIATED KINASE1 (BAK1) to the complex [64,65,66,67,68]. BR signals are eventually relayed to BRI1-EMS-SUPPRESSOR1 (BES1) and BRASSINAZOLE-RESISTANT1 (BZR1), two transcription factors that regulate the expression of BR-responsive genes [69,70,71]. The glycogen synthase kinase3-like kinase BRASSINOSTEROID INSENSITIVE2 is a negative regulator of BR signaling. It phosphorylates and inactivates BES1 and BZR1 by inhibiting DNA binding activity and stimulating degradation of the two proteins [69,70,72].

Recent studies have shown that FER is involved in BR-mediated plant growth and resistance. First, the expression of genes encoding FER and two other *Cr*RLK1L subfamily members (HERCULES1 and THESEUS1) are induced by BRs and modulated in BR mutants. The transcript levels of the three genes are upregulated in constitutive BR-response mutant *bes1-D* and are downregulated in the *bri1* mutant [73]. Furthermore, RNA interfering (RNAi) knockdown of *FER* in *Arabidopsis*, which dramatically reduces but does not completely eliminate the transcript level of *FER*, has no effect on the growth of seedlings in response to BRs under light [73]. However, complete disruption of FER results in severe hypersensitivity to 24-epibrassinolide (EBL) under light, suggesting that FER is a negative regulator of the BR response in light-grown hypocotyls [74]. In contrast, FER is required to promote BR responses in etiolated seedlings, because the hypocotyl growth of the *Arabidopsis fer* mutant is significantly less responsive than the WT upon treatment with different concentrations of EBL in the dark. This indicates the positive role of FER in the BR responses in dark-grown hypocotyls [74].

Recent findings confirm that FER is a receptor to both RALF1 and RALF23 [27,52,53,54]. The RALF–FER module has been shown to be involved in many different processes, including stress responses, cell growth, and immunity [54,75,76]. Among these functions, RALFs are involved in BR-regulated plant growth, and the two factors are generally functional in antagonism [77,78,79]. Specifically, overexpressing *RALF23* compromises brassinolide (BL)-induced hypocotyl elongation, and resulted in semidwarf plants with reduced root growth in *Arabidopsis* seedlings [77]. Moreover, the expression levels of *RALF23* decrease significantly upon BL treatment or in the constitutive BR-response mutant *bes1-D* [77]. The antagonism between RALFs and BL is also demonstrated in RALF1 situation. Overexpressing root-specific *RALF1* in *Arabidopsis* resulted in a decrease in the root length and number of lateral roots. However, upon treatment with brassinolide (BL), *RALF1*-overexpressing *Arabidopsis* plants display no increase in root length or the number of lateral roots compared to the WT [78]. Moreover, overexpressing *RALF1* promotes the expression of two BR biosynthetic genes, *CONSTITUTIVE PHOTOMORPHISM AND DWARFISM* (*CPD*) and *DWARF4* (*DWF4*) [80,81], both of which are downregulated by BR [78,79]. However, simultaneous treatment with RALF1 and BL represses expression of RALF1-inducible genes (e.g., the proline-rich protein-encoding genes *PRP1* and *PRP3*) [78], suggesting an antagonistic relationship between the BR and RALF signaling pathways. Given that FER is the RALF1 and RALF23 receptor, and the expression levels of *FER* and *RALF* can be modulated by BR [73,77], it is reasonable to extrapolate that FER plays a role in the RALF1 or RALF23-regulated BR response. However, the function of FER and its related partners involved in this process require further research to accurately elucidate.

In addition to plant growth, FER is also involved in BR-regulated heat tolerance in plants. Overexpressing the tomato *BZR1* promotes the production of apoplast H_2_O_2_ and enhances heat tolerance [82]. However, mutated *BZR1* compromises H_2_O_2_ production in the apoplast and heat tolerance by suppressing the induction of *RESPIRATORY BURST OXIDASE HOMOLOG1* (*RBOH1*) in tomatoes [82]. Moreover, silencing the tomato *FER* paralogs, *FER2/3*, also inhibits BZR1-dependent *RBOH1* induction, apoplast H_2_O_2_ production, and heat tolerance [82]. Further analysis showed that BZR1 binds to the *FER2* and *FER3* promoter to induce their expression, suggesting that BZR1 regulates RBOH1-dependent ROS production, at least in part through FER2/3 during heat tolerance in tomatoes [82].

## 5. Ethylene

Ethylene is a simple gaseous plant hormone that plays profound roles in plant life cycles. In addition to the well-known role in regulating fruit ripening [83], ethylene participates in plant cell division, seed germination, root hair formation, tissue differentiation, sex determination, and the responses to multiple biotic and abiotic environmental cues [84,85]. Ethylene is de novo synthesized from methionine by three enzymatic steps: methionine is converted to S-adenosyl methionine (SAM) by SAM synthase; then 1-aminocyclopropane-1-carboxylic acid (ACC) synthase (ACS) converts SAM into ACC, which is subsequently used to produce ethylene by ACC oxidase [84,85]. Recent findings reveal that FER is involved in ethylene-regulated cell growth [74] and fruit ripening [86,87].

FER transcript levels are promoted by increasing ethylene concentration, and FER is required for normal *Arabidopsis* growth in response to ethylene [74]. Loss of functional FER in *Arabidopsis* leads to inhibited ethylene-inducible hypocotyl growth compared to the WT (*Col*-*0*) [74]. Complementation assay by introducing the WT *FER* expression cassette into the *fer* mutant recovers inhibited hypocotyl growth, suggesting that FER plays a crucial role in the ethylene response [74]. Interestingly, although *FER* is ethylene-inducible and disrupted FER results in an aberrant ethylene response, overexpressing this gene in WT *Arabidopsis* does not affect the ethylene response, suggesting that as yet unknown limiting factor(s) might act downstream of FER in response to ethylene [74].

The FER-mediated ethylene signal also interacts with BR responses. FER is required for the BR response in etiolated seedlings [74]. AgNO_3_ is functional in reversing the hypocotyl-shortening defects normally seen in the dark-grown *fer* mutant [74]. When the WT and *fer* mutant *Arabidopsis* seedlings are treated in the dark in the presence of AgNO_3_, a higher concentration of brassinazole (BZR, a BR biosynthesis inhibitor) leads to severe hypocotyl shortening in the *fer* mutant compared to the WT [74]. Further investigation showed that in the presence of either high levels of EBL (100 nM) or no EBL, the hypocotyl length of *Arabidopsis* with disrupted ETHYLENE INSENSITIVE 2 (EIN2), which is required for ethylene signaling, showed no significant difference [74,88]. This indicates that inhibition of hypocotyl growth is largely dependent on ethylene biosynthesis and signaling. Furthermore, treatment with near-saturating levels of EBL stabilizes protein levels of ACS5 (ACC synthase 5), which catalyzes SAM into the ethylene precursor ACC [74]. Higher concentration of EBL stabilizes the ACS5 protein to promote ethylene production, but results in inhibited hypocotyl growth, suggesting that BR signals antagonize the hypocotyl growth of etiolated seedlings mediated by ethylene [74,89].

Given the critical roles of ethylene in fruit ripening, a growing body of evidence has demonstrated that FER is also involved in ethylene-regulated fruit ripening. Heterologous expression of apple *FER*-*like* 1 (*MdFERL1**)* and *MdFERL6*, two apple paralogs of *Arabidopsis FER*, inhibit ethylene production and delay fruit ripening in tomatoes (*Solanum lycopersicum*) [87]. Moreover, the two apple homologs physically interact with MdSAM synthetase [87]. Similar interactions have been reported in tomatoes and *Arabidopsis*. In tomatoes, SlFERL interacts with SlSAM synthetase 1, resulting in elevated SAM accumulation and ethylene production [86]. Thus, overexpression of *SlFERL* significantly accelerates the ripening of tomato fruit, whereas RNAi knockdown of *SlFERL* delays fruit ripening [86]. Moreover, the MADS-box transcription factor RIPENING-INHIBITOR (RIN), one of the master factors involved in ethylene-regulated fruit ripening, binds to the *SlFERL* promoter and activates its expression [86]. This finding suggests that SlFERL serves as a positive regulator during ethylene production and fruit ripening, which is contrary to the role of MdFERLs in ethylene production and fruit ripening. In *Arabidopsis*, FER interacts with SAM synthetase at the plasma membrane [90]. The *fer* mutant plants accumulate higher amounts of SAM and are much smaller than the WT in the presence of ethionine, a toxic analogue of methionine [90], suggesting that FER is a negative regulator of ethylene biosynthesis. Taken together, these data indicate that FER may function in a species-specific way during ethylene biosynthesis and signaling.

## 6. FER Plays a Pivotal Role in Regulating Plant Immunity

In natural environments, plants are continuously challenged by a large number of microbes, such as fungi, bacteria, and viruses. Thus, plants have evolved sophisticated immune systems to cope with the invasion of these microbes. Upon infection, some conserved molecules derived from microbes, including pathogen-associated molecular pattern (PAMP) and microbe-associated molecular pattern, are perceived and recognized by pattern recognition receptors (PRRs) to activate the immune system [91]. These PRRs are immune receptors localized on the cell surface to induce PAMP-triggered immunity and enhance basal resistance against invaders [91]. RLKs are the major PRRs that recognize PAMPs [92,93]. FER is one of the most well studied RLK members and plays critical roles in regulating plant immunity by participating in RALF signaling and resistance hormonal signaling pathways [5,54,75,93].

FER is a receptor of the RALF peptides [27,52,53,54]. SITE-1 PROTEASE (S1P) was reported to inhibit plant immunity by cleaving endogenous RALF propeptides, and FER is closely involved in this inhibition [52]. Stegmann et al. revealed that FER is a ligand that induces the formation of a complex composed of the receptor kinases EF-TU RECEPTOR (ERF) and FLAGELLIN-SENSING2 (FLS2), and their co-receptor BAK1, to activate plant immune signaling [52]. However, RALF23 negatively regulates immunity by reducing the formation of the ligand-induced ERF/FLS2–BAK1 complex in a FER-dependent manner [51,52]. In other words, FER positively regulates immunity because the *Arabidopsis fer* mutant plants show increased levels of bacterial (*Pseudomonas syringae* pv. *tomato* DC3000 coronatine-minus, *Pto* DC3000 COR) infection compared to the WT (*Col*-*0*) [52]. The latest finding reveals that FER regulates the plasma membrane nanoscale organization of FLS2 and BAK1, albeit in an opposite manner [51]. Moreover, RALF23 perception leads to rapid modulation of FLS2 and BAK1 nanoscale organization, which results in inhibited immune signaling in a FER-dependent manner [51]. As a closely related partner of FER, LRXs were also shown to regulate BAK1 nanoscale organization and immunity [51]. In the *lrx345* mutant, the formation of both the FLS2–BAK1 and ERF–BAK1 complexes was disrupted, suggesting that LRX3/4/5 are positive regulators in plant immunity [51]. In addition to RALF23, its closely related peptides RALF33 and RALF34 also negatively regulate immunity [52]. Given that the LRXs–RALFs–FER module plays a pivotal role in plant immunity, it is probable that FER is also involved in RALF33- and RALF34-regulated immunity in a similar way to RALF23.

However, some other research groups revealed that FER is a negative regulator of immunity. Keinath et al. reported that the *Arabidopsis fer* mutant in the *Ler* background is more resistant to bacterial infection than the WT (*Ler*), possibly because of the constitutively closed stomata in the *fer* mutant [94]. Moreover, the *fer* mutant plants are more resistant to powdery mildew infection, probably by inhibiting production of the fungal conidiophores [95]. Increased cell death and H_2_O_2_ production in the *fer* mutant plants, even in the absence of pathogens, could also contribute to enhanced resistance to the fungus [95]. Importantly, the fungal pathogen (*Fusarium oxysporum*) secretes a RALF peptide (Fusarium-RALF, F-RALF) that is homologous to plant RALFs [96]. F-RALF targets FER to induce extracellular alkalinization [27,96], which, in turn, activates a pathogenicity-related mitogen-activated protein kinase signaling cascade to facilitate hyphal growth and the virulence of the pathogen [97,98]. Fungus mutants with disrupted F-RALF fail to induce extracellular alkalinization and display significantly reduced pathogenicity to tomato plants, but elicit strong immune responses in host plants [96]. Moreover, the *Arabidopsis fer* mutant is more resistant against *F**. oxysporum* because the disrupted RALF–FER signaling pathway fails to induce extracellular alkalinization to facilitate pathogen survival [27,96]. The increased transcript levels of immune response marker genes, including *FLG22-INDUCED RECEPTOR-LIKE KINASE 1* (*FRK1*), *WRKY53*, and *PLANT DEFENSIN 1.2* (*PDF1.*2), in the *fer* mutant may also contribute to the enhanced resistance to *Fusarium* [96], suggesting that FER is a negative regulator of immunity. Taken together, the function of FER in immunity can be positive or negative, indicating its sophisticated roles in regulating plant immunity.

In addition to functioning as a RALF receptor, FER regulates immunity through hormones, including jasmonic acid (JA) and salicylic acid (SA). For instance, mutation of *FERONIA-like Receptor* (*FLR*) in rice enhanced the resistance to rice blast by increasing expression of *OsPR1a* (a pathogenesis-related protein, and a marker gene of the SA pathway) and *OsPR4* (a marker gene of the JA pathway) [99]. Moreover, FER was reported to regulate the antagonism between JA and SA to modulate immunity. Specifically, upon *Pto* DC3000 infection, bacteria-secreted phytotoxin coronatine (COR) structurally and functionally mimics the active form of JA, JA-isoleucine [100]. Then, COR utilizes MYC2, a master regulator in the JA signaling pathway, to activate transcription factor petunia NAM and *Arabidopsis* ATAF1, ATAF2, and CUC2 (NACs), including ANAC019, ANAC055, and ANAC072 [101]. These transcription factors subsequently regulate the expression of genes involved in SA biosynthesis and metabolism [101] (Figure 3). Among these NAC-regulated genes, *ISOCHORISMATE SYNTHASE 1* (*ICS1*), which is involved in SA de novo biosynthesis from isochorismate [102], was downregulated [101]. The SA glucosyl transferase gene 1 (SAGT1) is functional in converting SA into SA glucose ester (SGE) and SA O-b-glucoside (SAG) as an inactive storage form [103], and SA methyltransferase 1 (BSMT1) functions to convert SA into inactive methyl SA (MeSA) [104]. Transcription factors NACs could bind to the promoter of these two genes and increase their transcription [101]. Thus, COR functions through the MYC2–NACs module to inhibit SA accumulation, leading to compromised immunity [101]. However, FER interacts with and phosphorylates MYC2 to decrease its stability, resulting in increased immunity by disrupting the MYC2–NACs module-mediated inhibition of SA accumulation [105] (Figure 3). Moreover, RALF23 negatively regulates immunity by stabilizing MYC2 through FER [105], which is consistent with and explains the negative role of RALF23 in immunity reported by Stegmann et al. [52]. Interestingly, either overexpressing *RALF22*/*23* or disrupting FER or LRXs in *Arabidopsis* promotes both SA and JA synthesis by activating SA and JA biosynthetic genes, and results in increased SA- and JA-responsive genes, including *PR1*, *PR5*, *PDF1.2*, and *PDF1.3* [56]. It is well known that increased SA accumulation and expression of *PR*s contribute to enhanced immunity; however, this is contradictory to previous results that RALF is a negative regulator of immunity [51,52]. Additionally, JA and SA usually function antagonistically as SA is mainly associated with resistance to biotrophic pathogens, whereas JA tends to be associated with resistance to necrotrophs and herbivores. However, the results presented by Zhao et al. showed that overexpressing *RALF22*/*23* or disrupting FER or LRXs promoted the accumulation of both JA and SA [56]. All these data suggest a complicated role of the LRXs–RALFs–FER module in regulating SA/JA signaling pathways and plant immunity.

SA is the major resistance phytohormone that confers plant basal immunity to a broad range of pathogens. SA acts through its receptors nonexpressor of PR gene 1 (NPR1), NPR3, and NPR4 to regulate expression of pathogenesis-related protein (PR) genes to manipulate host immunity [106,107,108]. A large amount of evidence has demonstrated the critical role of SA signaling in pattern-triggered immunity (PTI), effector-triggered immunity (ETI), and the establishment of systemic acquired local resistance (SAR) in distal tissues [109,110]. However, the role of FER in SA signaling has not been investigated in detail. Zhao et al. revealed that disrupting FER or LRX*s* promotes the accumulation of SA by activating SA biosynthetic genes, including *ICS1*, *CALMODULIN BINDING PROTEIN 60g* (*CBP60g*), and *AVRPPHB SUSCEPTIBLE 3* (*PBS3*) [56]. Enhanced SA accumulation subsequently induces the expression of the SA response marker genes *PR1* and *PR5*, leading to enhanced immunity. In the presence of higher SA content, the cytosolic localized NPR1 monomers release from oligomers and translocate to the nucleus, where they function to promote the expression of downstream *PR* genes as a co-activator of transcription factor TGACG SEQUENCE-SPECIFIC BINDING PROTEIN 2 (TGA2) [111,112,113,114]. Thus, post-translational modifications of NRP1 mediated by kinase may also regulate plant immunity. For example, SNF1-RELATED PROTEIN KINASE 2.8 (SnRK2.8) phosphorylates NPR1 to promote its nuclear import, which results in enhanced expression of *PR*s and establishment of SAR in distal tissues [115].

## 7. Future Perspectives

Plant malectin-like receptor kinases, also known as *Cr*RLK1Ls, participate in many different plant processes, including cell growth, reproduction, abiotic stress responses, hormone signaling pathways, and immunity (Reviewed in [1,2,5,93]). FER is the most well studied member in the *Cr*RLK1L subfamily, and here we summarized recent advances related to the aforementioned functions. As a receptor kinase that is widely distributed in different plant tissues, FER interacts with multiple cellular signaling pathways by phosphorylating its partners, such as MYC2 [105], to modulate related responses. Nevertheless, the precise function of FER in many signaling pathways has not been well demonstrated, even with the identification of its interacting partners. For example, FER paralogs from *Arabidopsis* [90], apples [87], and tomatoes [86] have been confirmed to interact physically with the corresponding SAM synthetase. However, the biological function of the protein interactions has not been revealed, even with the genetic evidence that FER is closely involved in ethylene-regulated biological processes. Thus, it would be promising to identify the potential substrates and specific phosphorylation sites by performing a high-throughput quantitative phosphoproteomic analysis to decipher the mechanisms of FER in related processes.

In addition to modifying its partners, FER is also regulated transcriptionally or post-translationally by many internal and external factors. At a transcriptional level, *FER* is induced upon treatment with BR [73] and ethylene [74]. Moreover, the expression of *FER* is also controlled by many transcription factors. For example, BZR1 induces the expression of *SlFER2/3* by binding to the promoter of the two genes that mediate RBOH1-dependent ROS production and BR-regulated heat tolerance in tomatoes [82]. In addition, a MADS-box transcription factor RIN binds to the *SlFERL* promoter to activate its expression, which mediates ethylene-regulated fruit ripening [86]. At a post-translational level, FER is modified by multiple factors. For example, both RALF and ABA treatment promote phosphorylation and activation of FER [44]. Specifically, in the ABA signaling pathway, its negative regulator ABI2 interacts with and dephosphorylates FER to suppress its activity [44]. Moreover, the disruption of ERU decreases the accumulation of phosphorylated FER and increases the amount of phosphorylated H^+^-ATPase 1/2, which is involved in auxin-regulated root hair growth [27,31,32]. Thus, given the key functions of FER in plant growth and development, it is worthwhile investigating upstream factors involved in transcriptional and post-translational modifications of FER, which will provide new ideas for understanding FER-modulated cellular processes.

Immunity is a key weapon to fight back against invaders. FER plays different roles in response to different types of pathogens. FER loss of function improves resistance to powdery mildew [95], *F**. oxysporum* [96], but decreases resistance to *Pto* DC3000 COR^−^ [52], suggesting the contradictory roles of FER in regulating plant resistance. In addition, FER interacts with JA and SA to modulate immunity [56,105]. Bacteria-secreted COR utilizes MYC2 to activate transcription factor NACs, which are functional in inhibiting SA accumulation and immunity by regulating genes involved in SA biosynthesis and metabolism [101]. FER functions to suppress JA and COR signaling by phosphorylating and destabilizing MYC2, resulting in enhanced immunity [105].

ROS production is one of the common features in immunity responses, and FER is reported to regulate ROS production through the RopGEFs–RAC/ROPs–NADPH oxidases pathway to manipulate plant immunity [12,116,117]. However, FER is also involved in auxin- and ABA-mediated ROS production through the same pathway [12,43]; thus, a possible scenario is that ABA and auxin may also contribute to immunity through FER-dependent ROS production. Additionally, apoplastic pH, which is reported to be critical for plant cell growth, also plays a crucial role in plant immunity responses [31,118]. During infection, pathogens are able to sense, adapt, and alter the pH of microenvironment to facilitate their survival [118]. For example, upon establishment of infection of *Magnaporthe oryzae* in rice or barley, they will release ammonia to stimulate apoplastic alkalinization [119]. For plants, both ABA and auxin could regulate apoplastic pH through a FER-dependent pathway. Therefore, FER may also integrate ABA and auxin signaling to alter the apoplastic pH to regulate plant immunity responses [31,42,43]. Thus, determining the FER-regulated ROS production and apoplastic pH is a key step in determining the function of FER in immunity responses in plants.

Both JA and SA are important resistance hormones in plants. FER has been shown to be involved in the JA signaling pathway by interacting with master regulator MYC2 [105]; however, how FER is involved in SA signaling pathways has not yet been determined. Moreover, the LRXs–RALFs–FER module has been shown to play critical roles in immunity. Disrupting LRXs or FER promotes JA and SA biosynthesis by increasing their biosynthetic genes [56]. However, several key questions are unclear, i.e., how FER/LRXs regulate SA and JA biosynthetic genes, and whether FER modulates the SA downstream signaling pathway, perhaps through SA receptors (NPR1, NPR3, and NPR4) [106,108]. Thus, the key roles of FER in plant immunity merit further investigation.

## Figures and Tables

**Figure 1 ijms-23-03730-f001:**
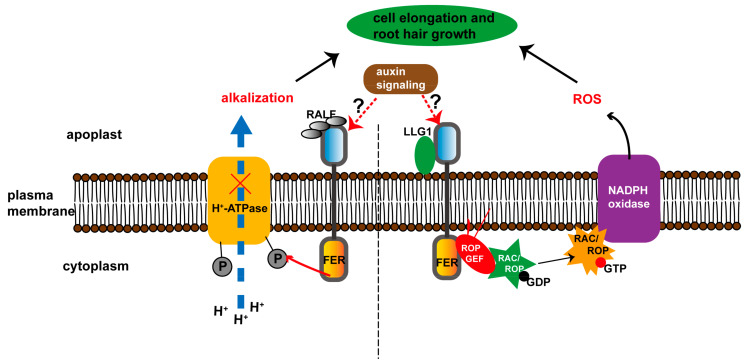
Schematic representation of the roles of FER in auxin-induced cell elongation and root hair growth. Two pathways have been reported to depict the roles of FER in auxin-induced root hair growth. In the first one, in the right side of the figure, FER regulates NADPH oxidase-dependent ROS accumulation through RAC/ROP to participate in auxin-induced root hair growth [12]. Specifically, the FER–ROPGEF module preferentially binds RAC/ROP in a GDP-dependent manner; upon signal activation, activated GTP-bound RAC/ROPs interact with NADPH oxidase to mediate ROS production, leading to root hair growth [12]. Additionally, LLG1 is a key component of the FER–ROPGEF–RAC/ROP signaling complex [24]. In the second pathway, on the left side of the figure, auxin regulates cell elongation by modulating the apoplast acidification in a FER-dependent manner. Specifically by binding its RALF ligand, FER induces phosphorylation of plasma membrane-localized H^+^-ATPase to inhibit proton transportation into the apoplast [27].

**Figure 2 ijms-23-03730-f002:**
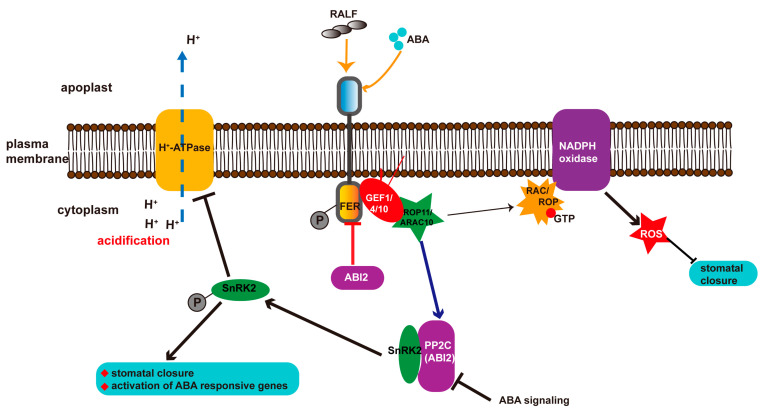
Schematic representation of the role of FER in the ABA signaling pathway. ABA directly represses ABI2 activity and releases activated SnRK2. Upon activation, SnRK2 induces expression of ABA-responsive genes and stomatal closure, inhibits H^+^-ATPase activity, and induces acidification of the cytosol, leading to inhibited cell growth. RALF and ABA increase the phosphorylation levels of FER, while FER activates ABI2 activity through the GEF1/4/10–ROP11/ARAC10 pathway, thereby inhibiting the ABA response [43]. FER also regulates ROS production through the GEF1/4/10-ROP11/ARAC10 pathway to regulate ABA-mediated stomatal closure [43]. ABI2 interacts with and dephosphorylates FER to suppress the FER–GEF–ROP pathway [44].

**Figure 3 ijms-23-03730-f003:**
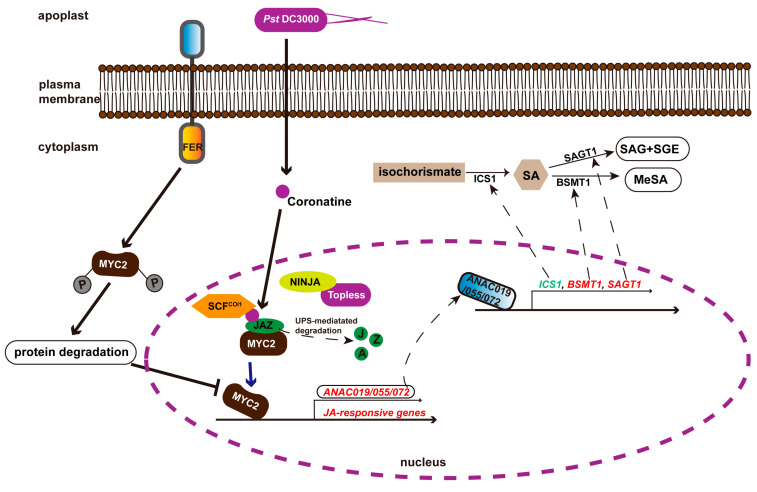
Schematic representation of the role of FER in regulating plant immunity through the JA and SA signaling pathways. Bacteria-secreted coronatine, which structurally and functionally mimics the active form of JA, utilizes MYC2 to activate the transcription factor NACs (ANAC019, ANAC055, ANAC072), which subsequently regulate the expression of genes involved in SA biosynthesis and metabolism to repress SA accumulation, and, thus, compromise plant immunity. FER regulates immunity by interacting with, phosphorylating, and destabilizing MYC2 to relieve the inhibitory effects of coronatine and JA on SA accumulation mediated by the MYC2–NACs module [73,101]. Red (BSMT1 and SAGT1) indicates activation, and green (ICS1) indicates suppression of transcription factor-regulated gene expression.

## Data Availability

Data is contained within the article.

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
