# Peer review of "FERONIA Receptor Kinase Integrates with Hormone Signaling to Regulate Plant Growth, Development, and Responses to Environmental Stimuli"

_ijms, 2022, doi:10.3390/ijms23073730_

Round 1

Reviewer 1 Report

The authors summarised recent advances in the functions of the receptor kinase FER, describing the different roles of FER in responding to different pathogen species and in regulating plant immunity, as well as the mechanisms by which FER is involved in the various signaling pathways of auxin, abscisic acid, brassinosteroids, ethylene, jasmonic, salicylic acid. The manuscript is well organised and comprehensive and reviews the latest scientific information, which represents a significant contribution to the field.

Author Response

We thank the reviewer's comments. According to the reviewer's comment, we just need to go through spell check of the manuscript. We have done this, please see the revised mansucript

Reviewer 2 Report

This is an interesting review on the interaction between FERONIA and endogenous phytohormones in regulation of the plant response to different environmental cues.   The manuscript generally reads very well although there are a few instances where some minor edits are recommended.

Abstract - phytohormones are critical chemicals (delete substrates).  Similarly on page 2 'phytohormones are critical plant chemicals' (delete substances).

Figure 2 caption - 2nd sentence - 'Upon activation...... and induces acidification of the cytosol..' 

Page 8 - half way down page 'shorter hypocotyl length compared to that of mock in etiolated.... '   Is there something missing here?

Page 9 1st paragraph - may consider including reference to Zhang et al. 2020 Plant Communications 1(4) RALF-FERONIA Signaling: Linking plant immune response with cell growth.

Page 10 - Paragraph 2 - SA is mainly associated with resistance to biotrophic pathogens whereas JA tends to be associated with resistance to necrotrophs and herbivores.  This mutual antagonism between SA and JA could be important when discussing the interactions between Feronia and SA/JA signaling pathways.  

Consider including reference to Yang et al 2020, J.Exp. Bot 17(6)  FER negatively regulates rice resistance to rice blast.  

Figure 3 - NINJA not NINJIA.  Please describe to roles of SAGT1 and BSMT1 and place in context within the text.  These could be important in regulation of SA levels in the cell.  

Page 12, line 3 - '...high-throughput  '

Page 12, paragraph 2, line 3 - 'induced upon treatment with..'

Page 12, paragraph 3 - suggest some discussion of hormonal crosstalk and its role in immunity.

Page 12, last paragraph - as above -do not ignore the important role of JA in plant immunity.

Author Response

Abstract - phytohormones are critical chemicals (delete substrates).  Similarly on page 2 'phytohormones are critical plant chemicals' (delete substances).

We thank the reviewer’s suggestion. We have removed this word.

Figure 2 caption - 2nd sentence - 'Upon activation...... and induces acidification of the cytosol..' 

We thank the reviewer’s suggestion. We have revised these spells.

Page 8 - half way down page 'shorter hypocotyl length compared to that of mock in etiolated.... '   Is there something missing here?

We thank the reviewer for pointing this out. We have corrected the statement.

Page 9 1st paragraph - may consider including reference to Zhang et al. 2020 Plant Communications 1(4) RALF-FERONIA Signaling: Linking plant immune response with cell growth.

We thank the reviewer’s suggestion. We have included this citation.

Page 10 - Paragraph 2 - SA is mainly associated with resistance to biotrophic pathogens whereas JA tends to be associated with resistance to necrotrophs and herbivores.  This mutual antagonism between SA and JA could be important when discussing the interactions between Feronia and SA/JA signaling pathways.  

Consider including reference to Yang et al 2020, J.Exp. Bot 17(6)  FER negatively regulates rice resistance to rice blast.  

We thank the reviewer’s suggestion. We have carefully discussed the SA and JA, we also included this citation.

Figure 3 - NINJA not NINJIA.  Please describe to roles of SAGT1 and BSMT1 and place in context within the text.  These could be important in regulation of SA levels in the cell.  

We thank the reviewer’s suggestion. We have corrected the typo in the revised figure 3. We also included the introduction of SAGT1 and BSMT1 in the text.

Page 12, line 3 - '...high-throughput  '

We thank the reviewer for pointing this out. We have corrected this word.

Page 12, paragraph 2, line 3 - 'induced upon treatment with..'

Revised.

Page 12, paragraph 3 - suggest some discussion of hormonal crosstalk and its role in immunity.

We thank the reviewer’s suggestion. We have included some discussion of the FER-mediated crosstalk of hormones and their roles in immunity.

Page 12, last paragraph - as above -do not ignore the important role of JA in plant immunity.

We thank the reviewer’s suggestion. We have included JA in the last paragraph.

Reviewer 3 Report

Receptor kinase-mediated signalling in plant cells is a fascinating and widely investigated field of research. Keeping pace with novel information in the field is clearly very important, and a large number of review articles have been published over the years to summarize recent developments and highlight possible new avenues for further research. The present manuscript focuses on FERONIA receptor kinases in connection with the action of some key plant hormones. The approach used here is very similar to the one used in a review article published a few years ago (Hongdong Liao, Renjie Tang, Xin Zhang, Sheng Luan, Feng Yu FERONIA Receptor Kinase at the Crossroads of Hormone Signaling and Stress Responses Plant and Cell Physiology, Volume 58, Issue 7, July 2017, Pages 1143–1150, mentioned in the manuscript). However, additional, more recent relevant literature is surprisingly overlooked, making this manuscript somewhat incomplete, whereas some other papers cited are a bit outdated A partial list of papers that should be considered is reported below:

Blackburn, M. R., Haruta, M., & Moura, D. S. (2020). Twenty Years of Progress in Physiological and Biochemical Investigation of RALF Peptides(1)( OPEN ). Plant Physiology, 182(4), 1657-1666. https://doi.org/10.1104/pp.19.01310

Kim, D., Yang, J. Y., Gu, F. W., Park, S., Combs, J., Adams, A., Mayes, H. B., Jeon, S. J., Bahk, J. D., & Nielsen, E. (2021). A temperature-sensitive FERONIA mutant allele that alters root hair growth. Plant Physiology, 185(2), 405-423. https://doi.org/10.1093/plphys/kiaa051

Li, C., Wu, H. M., & Cheung, A. Y. (2016). FERONIA and Her Pals: Functions and Mechanisms. Plant Physiology, 171(4), 2379-2392. https://doi.org/10.1104/pp.16.00667

Ortiz-Morea, F. A., Liu, J., Shan, L. B., & He, P. (2022). Malectin-like receptor kinases as protector deities in plant immunity. Nature Plants, 8(1), 27-37. https://doi.org/10.1038/s41477-021-01028-3

Solis-Miranda, J., & Quinto, C. (2021). The CrRLK1L subfamily: One of the keys to versatility in plants. Plant Physiology and Biochemistry, 166, 88-102. https://doi.org/10.1016/j.plaphy.2021.05.028

Zhang, X., Yang, Z. H., Wu, D. S., & Yu, F. (2020). RALF-FERONIA Signaling: Linking Plant Immune Response with Cell Growth. Plant Communications, 1(4), Article 100084. https://doi.org/10.1016/j.xplc.2020.100084

Zhu, S. R., Fu, Q., Xu, F., Zheng, H. P., & Yu, F. (2021). New paradigms in cell adaptation: decades of discoveries on the CrRLK1L receptor kinase signalling network. New Phytologist, 232(3), 1168-1183. https://doi.org/10.1111/nph.17683

In a difference to the mentioned review article by Liao et al., the manuscript under consideration is not very clearly written. The descriptions of the signalling modules are convoluted, and the schemes also contain too much information, without giving the reader the possibility of getting immediate grasp of essential information. Moreover, many awkward sentences are found throughout the manuscript. A few surprising mistakes can also be spotted (examples below):

Line 42 Madagascar periwinkle is not a scientific name. Writing it in italics shows that the authors are not aware of the difference between scientific names and common names.

Lines 49-50 The description of fertilization surprisingly does not take into account the obvious fact that Angiosperms have double fertilization, with 2 sperms.

Lines 93-94 “whether” is used in indirect speech, the question mark should not be used here.

Author Response

Receptor kinase-mediated signalling in plant cells is a fascinating and widely investigated field of research. Keeping pace with novel information in the field is clearly very important, and a large number of review articles have been published over the years to summarize recent developments and highlight possible new avenues for further research. The present manuscript focuses on FERONIA receptor kinases in connection with the action of some key plant hormones. The approach used here is very similar to the one used in a review article published a few years ago (Hongdong Liao, Renjie Tang, Xin Zhang, Sheng Luan, Feng Yu FERONIA Receptor Kinase at the Crossroads of Hormone Signaling and Stress Responses Plant and Cell Physiology, Volume 58, Issue 7, July 2017, Pages 1143–1150, mentioned in the manuscript). However, additional, more recent relevant literature is surprisingly overlooked, making this manuscript somewhat incomplete, whereas some other papers cited are a bit outdated A partial list of papers that should be considered is reported below:

Blackburn, M. R., Haruta, M., & Moura, D. S. (2020). Twenty Years of Progress in Physiological and Biochemical Investigation of RALF Peptides(1)( OPEN ). Plant Physiology, 182(4), 1657-1666. https://doi.org/10.1104/pp.19.01310

Kim, D., Yang, J. Y., Gu, F. W., Park, S., Combs, J., Adams, A., Mayes, H. B., Jeon, S. J., Bahk, J. D., & Nielsen, E. (2021). A temperature-sensitive FERONIA mutant allele that alters root hair growth. Plant Physiology, 185(2), 405-423. https://doi.org/10.1093/plphys/kiaa051

Li, C., Wu, H. M., & Cheung, A. Y. (2016). FERONIA and Her Pals: Functions and Mechanisms. Plant Physiology, 171(4), 2379-2392. https://doi.org/10.1104/pp.16.00667

Ortiz-Morea, F. A., Liu, J., Shan, L. B., & HeMalectin-like receptor kinases as protector deities in plant immunity, P. (2022). . Nature Plants, 8(1), 27-37. https://doi.org/10.1038/s41477-021-01028-3

Solis-Miranda, J., & Quinto, C. (2021). The CrRLK1L subfamily: One of the keys to versatility in plants. Plant Physiology and Biochemistry, 166, 88-102. https://doi.org/10.1016/j.plaphy.2021.05.028

Zhang, X., Yang, Z. H., Wu, D. S., & Yu, F. (2020). RALF-FERONIA Signaling: Linking Plant Immune Response with Cell Growth. Plant Communications, 1(4), Article 100084. https://doi.org/10.1016/j.xplc.2020.100084

Zhu, S. R., Fu, Q., Xu, F., Zheng, H. P., & Yu, F. (2021). New paradigms in cell adaptation: decades of discoveries on the CrRLK1L receptor kinase signalling network. New Phytologist, 232(3), 1168-1183. https://doi.org/10.1111/nph.17683

 We thank the reviewer’s suggestion. We have included these citations in the manuscript and also remove some outdated references.

In a difference to the mentioned review article by Liao et al., the manuscript under consideration is not very clearly written. The descriptions of the signalling modules are convoluted, and the schemes also contain too much information, without giving the reader the possibility of getting immediate grasp of essential information. Moreover, many awkward sentences are found throughout the manuscript. A few surprising mistakes can also be spotted (examples below):

Line 42 Madagascar periwinkle is not a scientific name. Writing it in italics shows that the authors are not aware of the difference between scientific names and common names.

We thank the reviewer for picking out this mistake. We have corrected this.

Lines 49-50 The description of fertilization surprisingly does not take into account the obvious fact that Angiosperms have double fertilization, with 2 sperms.

We thank the reviewer for picking this out. We have corrected this.

Lines 93-94 “whether” is used in indirect speech, the question mark should not be used here.

We thank the reviewer’s suggestion. We have corrected them.

We also polished our manuscript thoroughly and rewrote many awkward sentences to make it easier for readers to understand.

Round 2

Reviewer 3 Report

The authors improved the manuscript, but still there is some degree of inaccuracy in their revised version. Just two examples: 

Line 83: auxin-meidated

Line 527: "The most important hormone" cannot be two hormones. Which one is the most important? If they both are equally important, the sentence should be rephrased.

The references suggested in my previous comments were just thrown in, but not discussed. In the short time that occurred between receiving the decision letter and submitting the revised version, I guess the authors hardly had a chance of actually reading those articles. In general, I maintain my opinion that  the manuscript could be better organized and written, but I leave the Editor the choice whether it is worth publishing.

Author Response

Dear Reviewer:

        Thank you so much for your suggestions for improving the quality of our manuscript. In the new version, we have simplified all of our models, we also discussed the newly cited references, especially the one that suggesting the importance of feronia-temperature sensitive mutant in root growth. We also updated some new references and removed some outdated ones. Please see the newly revised version for detail. The blue color labels the newly revised section.

The authors improved the manuscript, but still there is some degree of inaccuracy in their revised version. Just two examples: 

Line 83: auxin-meidated

We thank the reviewer for picking out this mistake. We have corrected this one.

Line 527: "The most important hormone" cannot be two hormones. Which one is the most important? If they both are equally important, the sentence should be rephrased.

We thank the reviewer for picking this out. We have rephrased this sentence.

The references suggested in my previous comments were just thrown in, but not discussed. In the short time that occurred between receiving the decision letter and submitting the revised version, I guess the authors hardly had a chance of actually reading those articles. In general, I maintain my opinion that  the manuscript could be better organized and written, but I leave the Editor the choice whether it is worth publishing.

Round 3

Reviewer 3 Report

I appreciate the efforts made by the authors. The manuscript is now more informative than in its previous versions.